# Low Latency TOE with Double-Queue Structure for 10Gbps Ethernet on FPGA

**DOI:** 10.3390/s23104690

**Published:** 2023-05-12

**Authors:** Dan Yang, Xuhan Xu, Tianyang Chen, Yanhao Chen, Junjie Zhang

**Affiliations:** Key Laboratory of Specialty Fiber Optics and Optical Access Networks, Shanghai Institute for Advanced Communication and Data Science, Shanghai University, Shanghai 200072, China; yangdan123@shu.edu.cn (D.Y.); yixun@shu.edu.cn (X.X.); tyral_chen@shu.edu.cn (T.C.); yanhao_chen@shu.edu.cn (Y.C.)

**Keywords:** FPGA, TCP/IP offload engine, network traffic, low latency, model analysis

## Abstract

The TCP protocol is a connection-oriented and reliable transport layer communication protocol which is widely used in network communication. With the rapid development and popular application of data center networks, high-throughput, low-latency, and multi-session network data processing has become an immediate need for network devices. If only a traditional software protocol stack is used for processing, it will occupy a large amount of CPU resources and affect network performance. To address the above issues, this paper proposes a double-queue storage structure for a 10G TCP/IP hardware offload engine based on FPGA. Furthermore, a TOE reception transmission delay theoretical analysis model for interaction with the application layer is proposed, so that the TOE can dynamically select the transmission channel based on the interaction results. After board-level verification, the TOE supports 1024 TCP sessions with a reception rate of 9.5 Gbps and a minimum transmission latency of 600 ns. When the TCP packet payload length is 1024 bytes, the latency performance of TOE’s double-queue storage structure improves by at least 55.3% compared to other hardware implementation approaches. When compared with software implementation approaches, the latency performance of TOE is only 3.2% of the software approaches.

## 1. Introduction

In the information age, the network plays an increasingly important role in daily life and technological progress. Statistics show that more than 85% of the world’s networks use ethernet technology [1]. Among them, transmission control protocol/internet protocol (TCP/IP) is the most used communication protocol in ethernet that provides reliable transmission, on-time delivery, and traffic control [2].

In the past, when CPUs were used to process TCP/IP data, there was a significant “scissor difference” between the CPU performance growth rate and the data volume growth rate. CPUs could no longer directly cope with the network bandwidth growth rate, which became a bottleneck in network performance [3].

Based on this background, the use of dedicated hardware devices or processors for network data processing has become a popular research direction, and TCP/IP offload engine (TOE) technology has been proposed. TOE technology which implements TCP/IP engine offload technology has civil as well as commercial value. First, traditional NICs perform TCP/IP protocol-related operations by using software drivers, which will increase CPU thread consumption in terms of data replication, protocol processing, interrupt handling, etc. If the TCP/IP protocol is implemented purely in hardware, it will largely offload the CPU overload burden on the server. Secondly, if the TCP/IP protocol is designed and implemented in TOE hardware, it can be made into a dedicated chip, which can not only reduce the task of CPU data processing but also optimize the space volume of existing NICs and reduce the cost. Compared to traditional embedded and ASIC implementations, field-programmable gate array (FPGA) hardware implementations have the advantages of short design cycles and low research overhead. Therefore, FPGA-based TOE techniques are more commonly used.

With the development of computer interconnection networks, data center networks have emerged to provide a strong backbone for the rapid storage and efficient processing of massive data, becoming a core infrastructure for countries and enterprises [4].

A wide variety of applications are carried out within the data center network, and the usual data flow characteristics in private clouds can be divided into small data flows with explicit deadlines and large data flows that are not sensitive to latency [5]. Among them, small data streams account for a high number of distributed computing or web service requests, which often require minimal latency, while large data streams are distributed in a small number of relevant storage applications, which need to maintain a high throughput to meet the application requirements. In order to realize multiple session and low latency requirements for high-performance networks, it is very necessary to conduct research on TOE. Therefore, this paper conducts a study on the latency of TOE multiple TCP session data storage and transmission processing in a data center environment.

The rest of this paper is organized as follows. We discuss current research on hardware-based TOEs and low-latency TCP stack optimization in Section 2. Section 3 presents our proposed TOE reception transmission delay model. This section contains the introduction of TOE framework architecture and reception principle, parameterization of the model, and the analysis of factors affecting reception transmission delay. In Section 4, we describe solutions to problems that occur in design such as transmission scheduling and consistency. Section 5 presents our experiments and analysis of results to verify the functionality and performance of the TOE. Finally, in Section 6, we make a summary of this paper and a discussion of future work.

## 2. Related Works

TOE is a technology that has been developed for nearly two decades since its conception. In the academic field, Intel first conducted a theoretical modeling analysis of the use of TCP offload engine technology on end-to-end servers, which analyzed the feasibility of the technology and the improvement of performance [6]. In 2013, Langenbach U’s team compared the latency performance of three different platforms supporting 10-gigabit TCP/IP protocol stacks, including a common software stack, a stack based on software TOE with kernel bypass, and a hardware-TOE-based stack [7]. They concluded that the hardware solution has a significant advantage in latency performance by comparison. The system has been tested to have a minimum latency of about 5.5 us, which is particularly suitable for low-latency applications in short-range or tightly coupled networks. In 2015, SIDLER D’s team at Zurich Polytechnic implemented a more fully functional 10-gigabit TOE based on FPGAs using the high-level synthesis (HLS) logic design approach [8]. In 2016, their team implemented a TOE for low-latency data center networks by tailoring and optimizing the TCP stack functionality based on the previous design for the network characteristics of data centers [9]. In 2019, SIDLER D’s team conducted further research and introduced another 100 G FPGA-based design TOE called Limago, which supports TCP’s window expansion factor on top of the basic TCP protocol functionality making it possible to apply it to long fat networks with large bandwidth delay products [10]. Furthermore, several teams used the ethernet interface chip from WIZnet and implemented a relevant TCP offload application using the FPGA+W5500 architecture [11,12]. In 2020, Wu’s team at Central China Normal University implemented a 10-gigabit TOE with a single TCP session and fixed configuration as a TCP server side based on FPGA [13]. In 2023, Yang’s team proposed a TOE using an architecture that separates the data-sending channel from the receiving channel, with the protocol-sharing module unifying the management of session status and network communication events, which demonstrated a network latency performance of less than 1 μs for a TCP packet of 512 bytes [14].

In the industrial field, most of the TOE products currently used in high-end communication scenarios have partnerships with well-known FPGA design companies such as Xilinx, Actel, and Altera. The most famous ones include Intilop [15], Dini Group [16], PLDA [17], etc., which specialize in ultra-high bandwidth and ultra-low latency requirements for customers with IP licensing or complete FPGA-based NIC solutions.

Low-latency optimization for TOE has seen a lot of progress in recent years. Many research results have been generated at the hardware acceleration level. The most common optimization is the storage structure and the transmission path based on FPGA. For different TCP session numbers, the storage structure of TOE varies.

In the case of a small number of TCP sessions, an on-chip block random-access memory (BRAM) storage structure is generally used to store data. Ding’s team at Fudan University designed a TOE supporting a single TCP session based on HLS, where data are stored to BRAM and read out directly [18]. They constructed a bypass data channel with the operating system kernel, and the latency is reduced to one-tenth of the latency between a Linux host without TOE and a Windows host. SIDLER D’s team proposed the idea of latency optimization by moving the receive buffer from double data rate three synchronous dynamic random-access memory (DDR3) to BRAM, and the reception latency was halved from about 2.7 us to 1.3 us for a TCP packet payload of 1024 bytes [9]. Wu’s team used a time-division multiplexing design concept and a weightless polling scheduling algorithm to control the sequential reads and writes of DDR3 memory, and achieved a low latency of 1.466 us for a TCP packet payload of 1460 bytes [13]. Those manufacturers mentioned above usually support supporting a limited number of 64–128 TCP sessions, which is applied to high-frequency transaction scenarios.

In the case of a large number of TCP sessions, an external DDR3 storage structure is used instead of on-chip BRAM to store data. SIDLER D’s team used DDR3 to write and read data, which supported 10,000 concurrent TCP sessions of TOE [8]. The architecture can be designed to change the session lookup data structure and buffers as the number of sessions increases [8]. To meet the needs of latency-sensitive users and applications such as data centers, Xiong’s team uses multiple TCP session management pools and DDR3 storage architecture to support 1024 TCP sessions. They also proposed an over-send ACK mechanism to achieve an end-to-end latency of 4.2 us [19].

In order to meet the multiple TCP sessions and low latency requirements of data center networks, TOE optimization must maintain a balance between session number and latency performance, since each TCP session requires a certain amount of storage buffer for receiving and sending, and standard FPGAs have limited on-chip memory resources. Taking the current Xilinx Virtex-7 FPGA VC709 platform used in this paper as an example, it is configured with 1470 BRAMs of 36 Kb size, which means the on-chip storage BRAM resource is about 52 Mb. Assuming that each session is allocated a space size of 2 Mb as configured, the limited on-chip resources can only support a maximum of 26 sessions [20]. To achieve hundreds or thousands of high concurrent TCP session requirements, we must move the packet storage from on-chip BRAM to external DDR3. However, due to the initialization process and read/write data characteristics of DDR3 itself, the latency will be greatly increased.

In addition to this, the existing hardware solutions mentioned above use BRAM or DDR3 storage structure to store data, and require two conditions to be met before the data are read: the data are fully stored and the TCP state is ready for transmission. As a result, their transmission delays are not constant because of the variation in the payload length. If the payload length keeps increasing, the latency will also keep increasing, resulting in instability of latency performance. Furthermore, they are conducted under the assumption that the spatial processing rate of the server application layer exceeds the data sending rate, and do not consider and analyze the transmission latency when the application layer processing rate is slow and blockage occurs.

In order to solve the above problems, the low latency optimization of TOE should be improved. TOE’s reception delay is an important part of it, so in this paper, the main contributions are as follows:
We analyze the TOE transmission structure for 10-gigabit ethernet and build an end-to-end TOE transmission delay model. From the perspectives of theoretical analysis and experimental verification, the correctness of the model is confirmed.A double-queue storage structure combining first input first output (FIFO) and DDR3 is proposed, which is capable of dynamically switching transmission channels and achieving a minimum end-to-end transmission delay of 600 ns for 1024 TCP sessions. We also use a multi-mode method of updating address length to achieve consistency in data transmission.A non-blocking data transmission method for multiple-session server application layer reception is proposed. A handshake data query update mechanism with priority is used to obtain the amount of transferable data at the application layer and achieve efficient slicing and transmission of stored data.

## 3. TOE Reception Transmission Delay Theoretical Analysis Model

### 3.1. TOE Framework Architecture

TOE is an FPGA-based full hardware TCP/IP offload engine that relieves the pressure on the CPU to parse ethernet data and manage TCP sessions. TOE supports the standard TCP stack functions of TCP session’s link building and breaking, sliding window, sorting order, timeout mechanism, and traffic control, with the characteristics of multiple sessions and low latency. The TOE framework architecture is shown in Figure 1, which contains five major modules:
Tx engine, which is used to generate new packets to send to the physical (PHY) layer.Rx engine, which is used to process incoming packets to send to the application layer.TCP session management pool, also called TCP PCB BLOCK.TCP session state manager, which is used to switch and transfer TCP state.Tx buffer and Rx buffer, which are used to store data.

The architecture uses the AXI-Stream interface protocol for data transmission at both the application layer interface and the ethernet interface, and all external interfaces use common interfaces for easy portability. The application layer interface includes two types of interfaces: one is the query interface, which is used to query the amount of data to be transmitted and select the transmission path, and the is the data interface, which is used to transmit payload data of TOE and application data of the application layer.

An ethernet data frame comes from the bottom physical layer, which is converted from the network data byte stream to the AXI-Stream protocol bus format through the PHY layer and the media access control (MAC) layer. The IP encapsulation and decapsulation module is responsible for parsing this received frame, and the TCP checksum calculation and detection module will check it.

The Rx engine detects, filters, and stores the ethernet data frame, modifies the TCP session management pool registers according to the information of the received data, and updates the state management table. After that, it stores the received payload data in the Rx buffer. Afterward, it interacts with the upper application layer through the application layer interface.

The Tx engine receives the data and session information sent by the application layer through the application layer interface and stores the data temporarily in the Tx buffer. Then, it queries the session status of the TCP session management pool, determines the timing of data sending, and updates the corresponding registers and session status. After checksum calculation and packetization of data frames, it is sent to the network by the physical layer interface.

The TCP session management pool can store all register information for reliable TCP communication, mainly including receive registers, send registers, timeout registers, and congestion control registers.

TOE is a secure FPGA-based TCP/IP protocol hardware offload engine since TCP itself is a reliable and connection-oriented transport layer communication protocol. In addition to positive acknowledgment and timeout retransmission, TCP also uses header checksum and sort reassembly to ensure orderly and reliable data transmission. The TOE also uses the AXI-Stream protocol at both the application layer interface and the ethernet interface, which uses handshake signals such as “valid” and “ready” to ensure the security of transmission. The double-queue storage structure design which will be introduced in the next section is based on the original single-path structure with the addition of a FIFO structure. It uses a scheduling strategy to ensure correct data transmission, which will guarantee the security of the hardware design.

### 3.2. TOE Reception Principle

In this paper, we focus our analysis on the TOE reception process, including the Rx engine, the Rx buffer, and other related modules. The TOE reception principle is shown in Figure 2. When an ethernet data frame comes in, the TOE parses the data frame into two parts: the frame header and the payload data. The frame header is sent to the state management and the payload data are sent to the data storage. The data storage is a double-queue structure, consisting of a fixed-depth on-chip FIFO and an off-chip DDR3.

Ideally, when the application layer has sufficient reception space, the transmission process does not have a blockage. Previous hardware solutions in other literature use BRAM or DDR3 storage structure to store data, and require two conditions to be met before the data are read: the data are fully stored and the TCP state is ready for transmission. As a result, their transmission delays are not constant because of the variation in the payload length. In contrast, the proposed TOE with a double-queue structure in this paper uses FIFO and DDR3 to store data, and the data reading condition only needs to satisfy whether the TCP state is ready to read the transmission. Therefore, the TOE chooses to read the FIFO data when the application layer transmission is not blocked. Since the write port and read port of FIFO are separated for non-blocking transmission, there is no need to wait for the FIFO data to be stored and it can be read directly, so the latency remains relatively stable.

When the application layer has insufficient space and the transmission may become blocked, the TOE of the double-queue structure proposed in this paper will choose to read DDR3 data. Since DDR3 has only one set of data interfaces, the TOE must wait for DDR3 to finish storing all the data before it can read the data, so the delay will vary with the payload length in DDR3 storage.

Therefore, the latency is different for different transmission paths. When reading FIFO data transmission, the latency is a fixed value; when reading DDR3 data, the latency is a variable value.

### 3.3. Proposed FPGA-Based TOE Reception Transmission Model Structure

Figure 3 shows the structure of the proposed TCP double-queue storage structure reception transmission delay model with multiple sessions and low latency on FPGA. The point-to-point transmission process of TCP data packets is discussed with the following assumptions: no data frames are lost during the transmission process, TOE is server-side, Host A is the client, and they communicate with each other via the TCP protocol. The TOE receive delay consists of three parts: propagation delay, transmission delay, and processing delay.

Under the condition that the transmission environment of the experimental device remains constant and the network state remains stable, the impact of the propagation delay and processing delay is minimal. Therefore, in this paper, we mainly focus on the transmission delay. The transmission delay $T_{t}$ is defined based on the “head-in, head-out” measurement mode, where the starting point is the first byte of the packet after the ethernet packet is sent to the PHY and MAC layers, and the ending point is where the first byte of the application layer data leaves the TOE. $T_{t}$ may be affected by the length of the data frame, the storage method, the buffer queue capacity, and the data scheduling mechanism to produce variations. $T_{t}$ contains the following components:
(1)$$T_{t}=T_{parse}+T_{save}+T_{read}$$
where $T_{parse}$ is the TCP packet parsing and preprocessing delay, $T_{save}$ is the packet carrying payload data storage delay, and $T_{read}$ is the payload data reading and transmission delay to the application layer.

The double-queue storage is a two-channel storage structure that combines on-chip FIFO and off-chip DDR3, and one is selected to transmit TCP data to the application layer. The internal design structure of TOE consists of three modules: parsing and preprocessing, data storage, and data read. The data storage module consists of TCP session state management, direct storage, and indirect storage modules. The data read module consists of direct read control, indirect read control, and APP interface arbitration. FIFO storage is used as a partition between modules to reduce interference.

As can be seen from Figure 3, the entire TOE reception process is described as follows: Host A sends ethernet data to the TOE, which is converted from a network data byte stream to the AXI-Stream protocol bus format through the PHY layer and MAC layer. The parsing and preprocessing module implements protocol parsing of ethernet frames, dividing the TCP packet into two parts: the frame header and the payload data. The frame header contains the length of the frame, protocol type, destination, source IP, etc.

Based on the triple-redundant HASH algorithm lookup management table and management pool structure of our group’s gigabit TOE project, the TCP session state manage module performs HASH operations on different quaternion information (source IP address, destination IP address, source port, and destination port) to obtain multiple session TCP data information from the session connection resource pool [19]. Based on the session index and session status of the current received frame, it determines whether the current received TCP data frame is a sequence-preserving frame. If it is not, the current frame is discarded, and if it is a sequence-preserving frame, the current data are stored.

In this paper, in order to reduce the data reception delay, we store the payload data using two methods: one to the on-chip payload FIFO, and the other to the external DDR3 storage space. The FIFO stores the payload data and the last flag of the current packet, while DDR3 double-queues the payload data of all the 1024 session packets supported by this design, and the address of the DDR3 storage space is provided by the TCP session state manage module.

As shown in Figure 3, the data read module implements the function of delivering the data temporarily stored locally by the TOE to the application layer. In order to optimize the transmission efficiency between the TOE and application layer and reduce the delay caused by the coupling of transmission interface, the TOE and application layer designed in this article adopt the memory copy mode.

As shown in Figure 4, the TOE queries the remaining receiving space of the application layer through the APP query interface, then reads the temporary storage data of the TOE storage space, and finally moves the data to the receiving space of the application layer.

In order to adapt to the high concurrency and high throughput data transmission, the application interaction interface function designed in this paper, i.e., the APP query interface, is shown in Figure 5, where the TOE initiates the interaction request signal (req) and uploads the payload length (X) and the session (index) to be transmitted, and waits for the application layer’s response (ack) signal and feedback on the amount of receivable data (Y).

Taking two data packets sent by Host A in Figure 5 as an example, the payload data length of the first packet is 100 bytes and its session index is A. After TOE stores the payload data into FIFO and DDR3, the data read module initiates query interaction with the application layer using a handshake data interaction method of requesting query and updating. That is, TOE sets the request signal called req to a high level, and at the same time, it sends the session index signal and the length signal X to be transmitted to the application layer. Currently, the session index is A, and X equals 100 bytes. These two signals are effective during the period when the request signal is set to high.

After the application layer receives these three signals, it returns the response signal, called ack, and the amount of data it can receive in its receiving space, called Y. Currently, Y equals 350 bytes. The query interaction is valid only when the request signal req of TOE and the response signal ack of the application layer are set to high simultaneously. Then, the data read module compares the current size of X and Y. Since X equals 100, Y equals 350, indicating that X is smaller than Y, so the application layer space is enough to receive the payload data of the first package. Therefore, the payload data are read out from the TOE storage space and transferred to the application layer’s receiving space.

Similarly, for the second package, the payload data length is 200 bytes and the session index is B. The data read module again initiates query interaction with the application layer. At this point, the request signal req is set to high again, the session index is updated to B, and the length signal X is updated to 200 bytes while waiting for the response signal ack and the space capacity Y of the application layer. Since the payload data of the first packet are already received by the application layer, the space capacity Y needs to be subtracted by 100, then Y becomes 250 from 350. The data read module again compares the current sizes of X and Y, and finds that 200 is smaller than 250, indicating that the application layer space is still sufficient to receive the payload data of the second packet. Therefore, the payload data are read out from the TOE storage space again and transferred to the application layer’s receiving space.

To further reduce the data reception delay, the data read module follows the order of executing direct read control first, and then indirect read control. According to the comparison result of X and Y, the application layer is judged to handle the speed of receiving payload data, and the faster one uses the direct read FIFO method to transmit data. Otherwise, the indirect read DDR3 method is used. Among them, the direct query implements the selection of data reading path of the payload FIFO and DDR3 through APP query interface I, while the indirect query module implements the slicing of the read length of DDR3 through the APP query interface II. The process of the data read module is as follows:

The direct query module applies for interaction with the application layer and compares X and Y. If X ≤ Y, it informs the direct read module to read the payload FIFO and informs the indirect read control module not to read DDR3. If X > Y, it informs the direct read module not to read the payload FIFO, and informs the indirect read control module to conduct an indirect query to apply for interaction with the application layer again. Then, it compares the current interaction result X and Y, takes the smaller value as the slice length to read the current data of DDR3 in pieces, and repeats the indirect query operation. Finally, the current data of DDR3 are read. At this point, the data read operation is complete.

Among them, the direct query module’s method to inform the indirect query module and the transfer scheduling strategy of two transfer paths will be described in detail in Section 4.1. The read address of the DDR3 current data and the remaining length are updated using a multi-mode approach, which will be described in detail in Section 4.2.

In order to prevent conflicts arising from two groups of query interfaces and data interfaces competing at the same time, the query transmission interface arbitration module arbitrates their priorities separately. The query arbitration process is to select direct first, then indirect, while the data arbitration process is to select the one that comes first in order of reception.

For the reception management of 1024 TCP sessions in this design, TOE uses a TCP session management mechanism in the TCP session state manager module that combines the session management table and session management pool of our subject group’s gigabit TOE. The HASH result of the quaternion information (source IP address, destination IP address, source port, and destination port) of different TCP sessions is taken as the address of the session management table. The session management table stores the session index and the status of the current session. The session management pool stores all relevant register information to ensure TCP communication including receive registers, send registers, timeout registers, congestion control registers, etc.

When storing the data of 1024 TCP sessions in this design, the TOE designed in this paper has the same storage method for both small and large numbers of TCP sessions, both using load FIFO and DDR3 dual-path storage. The data of the payload FIFO are only used to transfer the current single session data, which will be cleared whether they are read or not, while DDR3 provides a 2 Mb receive buffer for each TCP session, which can be stored until it is overwritten when not read. DDR3’s different TCP session storage writing pointers follow a combination of base low address and high address, with the base low address being the low bit of the HASH result and the high address being the session index. This can ensure that data are stored in DDR3 and partitioned without conflict.

Further, when reading the data of 1024 TCP sessions, we adopt a session arbitration method with priority, as described in Section 4.3. The method arbitrates the 1024 TCP sessions in the priority order of event trigger > time polling > feedback query. Then, it interacts with the application layer through the APP query interface II to ensure the efficient management of 1024 TCP sessions.

The double-queue storage structure and the dynamic selection of the transmission path of this design greatly reduce the delay of TCP data transmission and enable efficient double-queue processing of multiple session packets, avoiding packet loss due to transmission blockage of network data and improving the reliability of TCP data transmission.

### 3.4. Delay Model Parameterization

In this paper, a parametric modeling analysis is performed for the transmission delay $T_{t}$ to explore the factors affecting the TOE transmission delay. In the FPGA system, the bit width of the TCP data frame received by TOE is 64 bits and the clock frequency is 156.25 MHz. After the clock domain and bit width conversion, the data bit width is $B_{d}$ and the clock frequency is F. It is known that the traditional TCP packet frame header consists of a 14-byte MAC frame header, 20-byte IP frame header, and 20-byte TCP frame header (without considering the TCP option field), so the length is a fixed value of 54 bytes, assuming the payload length of the data frame is l byte.

The packet parsing and preprocessing delay $T_{parse}$ consists of the delay of parsing the ethernet packets sent by Host A to obtain the header information and payload data of the current ethernet packet. Since it is a pipeline architecture process, not affected by the length change, the delay is constant and noted as $C_{l}$. Then, the packet parsing and preprocessing delay $T_{parse}$ can be expressed by the following Equation (2):
(2)$$T_{parse}=C_{l}$$

The data storage delay $T_{save}$ varies depending on the storage method and needs to be discussed in two cases:
Direct storage latency. This contains the latency of the TCP session state manager module for state switching of the current TCP session, updating the state FIFO, and feeding the payload data into the payload FIFO before conducting direct queries. Since the direct query state machine determines that the condition for opening a query is that the state FIFO and the payload FIFO are non-empty, the latency of direct storage does not vary with variables such as payload length and is a constant noted as $C_{w1}$. The direct storage latency is expressed by the following Equation (3):
(3)$$T_{save1}=C_{w1}\;$$Indirect storage latency. Constant latency includes TCP session state manager processing latency, payload data waiting storage latency, and updating the finished FIFO’s latency after the completion of storage, noted as $C_{w2}$. Non-constant latency includes DDR3 write operations. Using the AXI interface protocol for burst operation to control the MIG IP core to write to DDR3 requires the write address, write data, and write response operations, which will be affected by the payload length and DDR3’s own characteristics. Ideally, when continuously writing payload data with a data bit width of $B_{d}$ bits and a length of l byte, the indirect storage latency can be expressed by the following Equation (4), where “⌈ ⌉” is the upward rounding sign:
(4)$$T_{save2}=\frac{\left\lceil 8l/B_{d}\right\rceil }{F}+C_{w2}$$

The data read time delay $T_{read}$ determines whether to slice or not based on the direct query result and thus uses different processing methods, which are discussed in two cases:
When the direct query results in X ≤ Y, i.e., the application layer receives data without blocking, then the data should be read directly. Thus $T_{read}$
includes direct query, notification of indirect query, reading the payload FIFO, and interface arbitration delay. Among them, the direct query time uses a handshake interaction, which mainly depends on the application layer response time, while other parts are pipeline architectures whose delay is denoted as $C_{r1}$. It assumes that the query request is initiated and waits $N_{c1}$ clock cycles for the application layer to respond to the ack signal. Therefore, the data read delay $T_{read1}$ is expressed by the following Equation (5):
(5)$$T_{read1}=\frac{N_{c1}}{F}+C_{r1}$$When the direct query results in X > Y, i.e., the application layer is slow to transfer data, then the data should be read indirectly. Thus $T_{read}$ includes indirect query, reads the DDR3, and interface arbitration time. In the case of indirect query, if the application layer feeds back the amount of transferable data Y = 0, it still needs to wait for the next round of queries until Y is not zero before it can start slicing and reading. If the application layer responds to the non-zero Y ($N_{c2}\geq N_{c1}$) after $N_{c2}$ clock cycles, and other parts are pipeline architectures whose delay is denoted as $C_{r2}$, the data read delay $T_{read2}$ is expressed by the following Equation (6):
(6)$$T_{read2}=\frac{N_{c2}}{F}+C_{r2}$$

In summary, according to the direct query results, the TOE transmission delay $T_{t}$ is divided into two cases, direct transmission delay $T_{t1}$ and indirect transmission delay $T_{t2}$, which can be expressed by the following two equations, where $C_{1}$ and $C_{2}$ are constants:
X ≤ Y:
(7)$$\begin{array}{ll} T_{t1} & =T_{parse}+T_{save1}+T_{read1}\\ & =C_{l}+C_{w1}+\frac{N_{c1}}{F}+C_{r1}\\ & =\frac{N_{c1}}{F}+C_{1} \end{array}$$X > Y:
(8)$$\begin{array}{ll} T_{t2} & =T_{parse}+T_{save2}+T_{read2}\\ & =C_{l}+\frac{\left\lceil 8l/B_{d}\right\rceil }{F}+C_{w2}+\frac{N_{c2}}{F}+C_{r2}\\ & =\frac{\lceil 8l/B_{d}\rceil +N_{c2}}{F}+C_{2} \end{array}$$

The performance improvement rate *R* for direct transmission delay compared to indirect transmission delay can be expressed in Equation (9):
(9)$$R=\frac{T_{t2}-T_{t1}}{T_{t2}}=\frac{\lceil 8l/B_{d}\rceil +\left(C_{2}-C_{1}\right)F+\left(N_{c2}-N_{c1}\right)}{\lceil 8l/B_{d}\rceil +C_{2}F+N_{c2}}$$

### 3.5. Analysis of Factors Affecting TOE Transmission Delay

From Equations (7) and (8), the TOE direct transmission delay $T_{t1}$ remains unchanged, while the TOE indirect transmission delay $T_{t2}$ changes with factors of the payload data length l, data bit width $B_{d}$, clock frequency F, DDR3 read/write characteristics, application layer processing data speed, and state machine processing fixed delay, which are analyzed below.

#### 3.5.1. Data Configuration Parameter Factors

When the data bit width $B_{d}$ and clock frequency F are constant, if the payload data length increases, the indirect transmission delay $T_{t2}$ increases in a stepwise manner. When the payload data length l is constant, if the data bit width $B_{d}$ or clock frequency F increase, the delay decreases. For 10 G ethernet data packets with a clock frequency of 156.25 MHz and a data bit width of 64-bit, pre-processing such as increasing the data bit width and clock frequency is required to make the data stream faster for storage and reading, etc.

#### 3.5.2. DDR3 Read/Write Characteristic Factor

Since DDR3 has only one set of data access interfaces, it cannot meet the demand of read-while-writing during high-speed 10 G ethernet data transmission. There will be a situation in which the previous data are not finished being read while new data are ready to be stored. These data need to compete for data access interface, which increases the latency. The AXI bus protocol used in this DDR3 read/write design is based on burst mode in a transaction, which reduces the address channel occupancy and improves the data transfer efficiency. However, the 4 k boundary rule of the AXI bus protocol also causes DDR3 to increase the number of bursts during the read/write process, thus increasing latency.

#### 3.5.3. Application Layer Processing Data Rate Factor

The application layer processes the payload data sent by the TOE to the corresponding process on the target port. The application layer response time, the buffer capacity capable of receiving TCP data, and the size of the data processing speed all affect the TOE query interaction results and thus affect the TOE indirect transmission latency. Suppose the initial free capacity of the application layer buffer is B bytes, at every $t_{s}$ time, the application can process $B_{s}$ byte data and release $B_{s}$ byte space capacity. Then, the application layer processes data at a rate of $8B_{s}/t_{s}$ bps. At time t, the total length of payload data that TOE has transmitted to the application layer is $l_{s}$ byte, and after receiving a new frame of TCP packets, the remaining length of payload data to be transmitted is $l_{t}$ bytes. During the direct query interaction, the amount of data requested for transmission X by TOE and the feedback on the amount of receivable data Y by the application layer can be expressed as Equations (10) and (11):
(10)$$X=l_{t}$$
(11)$$Y=B+\lfloor \frac{t}{t_{s}}\rfloor \times B_{s}-l_{s}$$

Therefore, when the response time, initial capacity, and processing data speed are constant, at different moments t, both the length of the TOE payload to be transmitted and the amount of data that can be transmitted at the application layer change, and according to Equations (7) and (8), the read transmission is selected between the payload data FIFO and DDR3 and the TOE transmission delay changes.

When the response time becomes shorter, the initial capacity becomes larger and the data processing speed becomes faster. TOE will not wait for the completion of the storage of DDR3 and directly read the payload data FIFO and send them to the application layer; thus, the TOE transmission delay is lower.

On the contrary, when the response time becomes longer, the initial capacity becomes smaller and the data processing speed becomes slower, making too much data pile up in DDR3, and the length of the payload to be transmitted exceeds the capacity of the application layer. It is thus necessary to wait for the next release time $t_{s}$ before transmission; thus, the TOE transmission delay is larger.

#### 3.5.4. State Machine Processing Fixed Latency Factor

According to Equations (7) and (8), if the state machines of the TOE system modules process the information more slowly, the TOE transmission delay will become higher. For example, the selection of the channel that needs to be transmitted after the TCP payload data storage is completed, the process of reading the data to be sent to the APP data interface, the storage of multiple TCP session data, and the recording of the reading address and the remaining payload length all need to be scheduled and updated in a timely manner.

## 4. TOE Transmission Structure Design Key Factors

In order to achieve efficient transmission of TCP data and reduce transmission latency, TOE’s double-queue storage transmission structure involves transmission scheduling, consistency problems, and the difficulty of handling data from multiple session connections. The design ideas are described as follows.

### 4.1. Transmission Scheduling Strategy

TOE receives new packets and cannot guarantee that its payload data are transferred from only one of the stored FIFOs and DDR3s to the application layer. If transmission scheduling is not performed, the data read module has problems such as duplicate reading and transmission packet loss, and cannot guarantee the uniqueness of the payload data transmission path, resulting in lower data transmission accuracy and higher latency.

As shown in Figure 6, this paper implements the transmission scheduling strategy by using channel flag and direct length.

The direct query and indirect query modules each maintain their own state machines, and the direct query has a higher priority. A channel flag value of 1 indicates that the data from the payload FIFO of the direct storage module are ready for transmission, and if it is 2, the DDR3 data from the indirect storage module are ready for transmission. The direct length is used by the multi-mode update address length module to locate the read pointer for the DDR3 payload data.

The transfer scheduling strategy uses three dual-port RAMs, where RAM1 and RAM2 each have remaining session transfer payload lengths, and RAM3 is used to store read pointers to the DDR3 payload data.

The design of the transmission scheduling strategy in this paper is as follows:

In the idle state, the direct query module waits for the non-empty signal of the payload FIFO and status FIFO and extracts the session index and payload length X. Then, it starts an interactive query to obtain a transmittable length Y. After reading the remaining length X’ stored in length RAM1, it judges:
If X ≤ Y and X’ = 0, assign a direct length to X and a channel flag to 1.If X > Y or X’ ≠ 0, assign a direct length to 0 and a channel flag to 2.

The two signals are stored to the flag FIFO. Then, the direct read module reads the payload FIFO data and judges the channel flag; if it is 1, the data will be transferred to APP data interface I, and if it is 2, the data will be discarded.

Secondly, the indirect read control module reads the two values of the flag FIFO and judges the channel flag:
If the channel flag equals 1, the indirect read control module writes the direct length value to the direct report FIFO and returns to an idle state.If the channel flag equals 2, this module reads the length of RAM2 to obtain the remaining length X’ and applies to interact with the application layer to obtain Y. Then, it takes the smaller value of X’ and Y as the slice length and reads the address of RAM3 to obtain a read pointer. Finally, this module reads the payload data stored in DDR3 in segments and transfers them to APP data interface II.

Cyclic interactive query and segmentation transfer operation until the current session remaining length X’ of length RAM2 is zero, end the transfer and return to an idle state.

### 4.2. Data Transmission Consistency

Since DDR3 stores all packets received by TOE carrying payload data, indirect transmission of DDR3’s current payload data storage location, i.e., read pointer or remaining length, is not updated, which can lead to DDR3 data reading errors and increase latency. Therefore, this paper designs multi-mode update address and length methods, respectively, to make TCP payload data consistent from input to output.

Figure 7 shows the multi-mode update address structure—three groups of update interfaces using round-robin scheduling arbitration out of a group of interfaces to update the address RAM. The address is the session index, and the data are the DDR3 data read pointers.

In this paper, three modes are designed to trigger the address update operation:
The first mode is the chain-building mode, i.e., the current TCP connection is established, the session index and DDR3 write address assigned by the previous module are obtained, and it applies to update the read pointer to the DDR3 write address.The second mode is the direct mode, indicating that the DDR3 read data pointer needs to skip the data transmitted by the direct channel. When the status information and direct report FIFO is non-empty, it obtains the current session index, reads the direct length, and applies to add the read pointer to the direct length.The third mode is the slicing mode, indicating that DDR3 has already transferred a part of this payload length, which obtains the session index and slice length of the current connection and applies to add the read pointer to the slice length.

The multi-mode update length method works on both frame lengths RAM1 and RAM2 which store the remaining length of the payload data. Taking length RAM2 as an example, as shown in Figure 8, unlike the update address method, the remaining length will be cleared when building a chain. When the current session data cannot go through the direct channel but are transmitted through the indirect channel, the remaining length will be added to the current payload length. If the DDR3 slice length data are read out, the remaining length will be subtracted from the slice length.

This design of multi-mode update address and length module ensures that when transferring through the indirect channel, the TOE can accurately locate the current storage location of the data to be transferred and the length to be read each time in the DDR3.

### 4.3. Multi-Session Priority Arbitration

When reading the data of 1024 TCP sessions, it will choose different reading methods according to the interaction results of the direct query module. When the application layer transmission is smooth, i.e., when the direct read module works, the payload data of the current TCP session will be read from the payload FIFO and transferred to the application layer. Then, it will loop to wait for the arrival of a new TCP session packet without affecting either one.

Otherwise, when the application layer transmission is not smooth, i.e., when the indirect read module works, TOE needs to use DDR3 data with slicing. If the data of the current TCP session has not been completely read, and now a new TCP session packet is received and waiting for being read, it is necessary to design arbitration mechanism to choose the order of reading. If not, it will lead to data-reading confusion. Furthermore, it will cause TOE to fall into a scenario where there is not enough space for the current single session transmission, resulting in blocked packet loss. Finally, the reception latency of TOE will increase a lot.

Therefore, this design then adds a multiple TCP session transmission method in the indirect read control module as shown in Figure 9. The design idea is to process new TCP session data first, then process the remaining uncompleted TCP session data of the system, and finally process the current TCP session data.

The specific design approach is to use a combination of three modules: event trigger, feedback query, and time polling, where the event trigger module contains the chain-building mode and direct mode described in Section 4.2, and the feedback query module contains the slicing mode. In this design, the priority order is event trigger > time polling > feedback query, and the length to be transmitted of different TCP sessions is stored in RAM2 with the address of the TCP session index. This design arbitrates the current TCP session index and the length to be transmitted, then it interacts with the application layer through the APP query interface II. After interaction, it chooses the reading mode according to the interaction result and selects one of the payload FIFOs’ or DDR3s’ slice readings to transmit data to the application layer, ensuring efficient management of 1024 TCP sessions.

Taking the new TCP packet with session index i and payload length X as an example, currently, the event trigger module obtains the channel flag of the flag FIFO; if it is 1, it transmits directly, if it is 2, it transmits indirectly. When direct transmission works, these three modules just wait without working. When indirect transmission works, the event trigger module updates the remaining pending transmission length of session index i stored in RAM2, i.e., it adds X. Then, it sends the session index i to the event FIFO and the updated pending transmission length (X + X_i_) to interact with the application layer.

The feedback query module locks session index k, which equals i, and obtains the application layer feedback length Y_k_. Now, the stored pending transmission length X_k_ equals X + X_i_. At this point, Y_k_ < X_k_, indicating that only part of the payload data can be transmitted, then the session index k is written to the feedback FIFO, and TOE interacts with the application layer according to the session index k and the updated pending transmission length X_k_–Y_k_. The interaction is cycled until the current pending transmission length is zero.

The time polling module polls at the B side of the dual-port RAM2, which is addressed with TCP sessions from 0 to 1023. The read data of RAM2 are cyclically queried to see if they are zero. If the data length of session index j is non-zero, it indicates that there are still pending data to be transmitted. Then, session index j is written to the polling FIFO and the read data X_j_ apply to interact with the application layer.

The multi-session priority arbitration module judges whether the three FIFOs are non-empty, and reads the stored TCP session data for the non-empty FIFOs in the priority order of event trigger > time polling > feedback query. Then, it obtains the arbitrated session index and the pending transmission length and interacts with the application layer.

Therefore, this method prioritizes the TCP session processing of new packets, then the session processing of polling for the presence of pending data, and finally the session processing of feedback queries for the presence of pending data. It can receive the TCP session data stream without conflict, avoid the situation where a single TCP session is stuck due to the inability of the application layer to receive data, and ensure the efficiency of a large number of TCP session data transmissions.

## 5. Experimental Design and Analysis of Results

To verify the functionality and performance of the TOE, a point-to-point platform was built and tested. The test connection diagram is shown in Figure 10, and they are connected using single-mode fiber and optical modules.

The TOE, which is server-side, is built on the XILINX VC709 development board, which is equipped with a Virtex-7 XC7VX690T, two DDR3 SODIMMs of 4 GB each, and four 10 G network interfaces, one of which is used.

Host A, which is client-side, using the WIN10 platform, is equipped with a commercial 10-gigabit NIC EB-SFP10G599-1F.

This paper uses Xilinx’s Vivado 2018.3 software as the development environment, Modelsim as the waveform simulation tool, Wireshark as a network packet capture software, a network program written in Python, and a network debugging assistant to construct TCP packets. We capture the input and output interfaces and internal debug signals of TOE using Vivado’s waveform and use the ILA kernel to measure the number of clock cycles elapsed in the running FPGA to calculate the latency.

In addition, the point-to-point network environment analyzed in this paper is usually at a very low packet loss rate, so the transmission delay caused by the packet loss is not discussed in this paper.

It is known that the data interfaces from MAC, TOE, and the application layer are different, and the system parameters are configured in Table 1:

### 5.1. TCP Data Transmission Validation Experiment

In order to verify the validity of data transmission, a TCP packet reception simulation test is conducted using Modelsim. Host A initiates a TCP link-building request and sends TCP data packets carrying a payload length of 64~1024 bytes to TOE after three handshakes. We take the first packet carrying a 64-byte payload as an example.

The blue part of Figure 11 shows the content of the payload data, with the low byte of the data increasing from 01 to the high byte of 40. The input data interface is in the AXI-Stream protocol format, as shown in Figure 12, with a total length of 118 bytes, including 54 bytes of frame header and 64 bytes of payload data.

After parsing and pre-processing, as shown in Figure 13, the data are stored in the payload FIFO of direct storage and the DDR3 memory of the indirect storage module, respectively. The high bit of the payload FIFO carries the last signal of the current frame.

The direct query module interacts with the application layer through the APP query interface I. As shown in Figure 14, at this time, the amount of data requesting to be transmitted X = 64, and the application layer feeds back the transmittable data amount Y = 4096; if X < Y, then direct transmission is possible. Therefore, the direct query module writes direct length = 64 and channel flag = 1 to the flag FIFO. The direct read module reads the payload FIFO and converts it into an APP data interface to send out.

It continues transmission until the t1 moment, as in Figure 15, and the direct query module continues to query when X = 1024, Y = 512, X > Y, the direct length equals 0, and the channel flag equals 2. It informs the indirect query module to continue indirect transmission through the APP query interface II. At the t2 moment, the indirect query X = 1024 and the application layer’s amount of receivable data Y’ has been updated to 4608, so X > Y’. Therefore, the indirect read module reads DDR3’s 1024 bytes of corresponding address data, converts them into the APP data interface II, and sends them to the application layer.

In summary, we can see that TCP data are transmitted according to the designed structure, the payload data content is consistent, and the data validity is verified correctly.

### 5.2. TOE Transmission Latency Performance Experiments

To test the communication transmission delay performance of TOE, two scenarios are set for analysis—Host A sends 1500 TCP packets with payload lengths of 64~1024 bytes and 1024 session numbers to TOE at an interval of 1 s, among which small packets with payload lengths less than 200 bytes are set with 80% weight. The number of clock cycles from the TOE input to output is counted, which is used to calculate the received transmission delay and compare the TOE transmission delay performance.

Scenario 1 is the ideal situation where the application layer bandwidth is sufficient and the transmission is unblocked. Compare the transfer latency of four storage modes:
Double-queue storage structure TOE.Single DDR3 storage structure TOE which has modified the internal logic so that direct query still selects DDR3 transfer.BRAM storage structure of other literature.Single DDR3 storage structure of other literature.

Scenario 2 is the actual situation. Modify the application layer processing data parameters, i.e., initial idle capacity, release time, released space capacity, and response time, and observe the transmission channel of the TOE as well as the delay variation characteristics.

#### 5.2.1. Application Layer Unblockage Scenario Latency Experiment

Figure 16 shows the comparison of TOE transmission latency with the existing hardware approaches for the application layer transmission unblockage scenario.

It can be found that when the TCP packet payload length is 64 bytes, refs. [9,21] all use the BRAM storage structure, the latency of which is about 700 ns. Ref. [8] uses the DDR3 storage structure, the latency of which is about 1.21 us. In comparison, the latency of the proposed double-queue storage structure TOE is an almost fixed value, which is 600 ns. From the above experimental results, when the payload length is 64 bytes, the latency performance of the proposed TOE improves by 14.3% compared to the BRAM structure and by 50.4% compared to the DDR structure.

As the payload length increases to 1024 bytes, the latency of the BRAM structure in [9,21] increases to about 1.344 us, while the latency of the DDR3 structure increases to almost 2.62 us. It can be found that when the TCP packet payload length is 1024 bytes, the latency performance of the proposed TOE improves by 55.3% compared to the BRAM structure and by 77.1% compared to the DDR3 structure.

It can be verified from Section 3.2 that the existing approaches of other hardware implementation approaches use BRAM or DDR3 to store data and need to wait for the completion of data storage before reading, so the latency varies with the payload length. In this paper, we propose a double-queue storage structure TOE, including both on-chip FIFO and off-chip DDR3 storage methods. The latency is a fixed value when the application layer is not blocked.

However, if TOE only uses a DDR3 storage structure without the FIFO storage method, the latency is about 1.3 us when the payload length is 64 bytes. According to Equation (10), compared with the double-queue storage structure, the performance improvement rate R of the TOE is 53.79%. When the payload length increases to 1024 bytes, the latency increases to about 1.75 us due to the delay of DDR3 data storage, and the performance improvement rate of TOE increases to 65.78%. It can be found that the latency of TOE varies with the payload length when using the DDR3 storage structure to send data, which also verifies the theoretical basis of Section 3.2 and the derivation of Equation (9) in Section 3.4.

Compared with the existing software approaches, it can be observed that the latency of the TOE double-queue structure is only 3.2% of the software implementation approach, which shows a transmission latency of about 18.5 us to 22 us [22].

Figure 17 shows the transmission latency of the double-queue structure and the single DDR structure for 1500 packets obtained statistically. As the index of packets increases, the transmission latency of the double-queue structure is stable at about 600 ns. The latency model of Equation (8) in Section 3.4 is verified. The transfer latency of the single DDR3 structure fluctuates which is affected by the DDR3 write data characteristics. It increases with increasing payload length and the latency is about 2~3 times that of the double-queue structure.

In summary, the double-queue structure TOE has better transmission latency performance than the BRAM structure, the single DDR3 structure, or even software TCP transmission. The latency characteristics are double-queue < BRAM < DDR3 < software TCP. As the TCP packet payload length increases, the delay performance improvement of double-queue structure TOE is more obvious. When the TCP packet payload length is 1024 bytes, the latency performance of the proposed double-queue structure TOE improves by 55.3% compared to the BRAM structure and by 77.1% compared to the DDR structure.

#### 5.2.2. Application Layer Blockage Scenario Latency Experiment

When the application layer transmission data rate changes, there will be data reception blockage. The initial conditions are set: initial free capacity is B = 2048, response period $N_{c2}$ =1, release space capacity $B_{s}$ = 2000, and update time interval $t_{s}$ = 5 us.

The transmission rate is changed by changing the release space capacity $B_{s}$ and update time $t_{s}$, respectively, and the transmission delay is observed and compared between the double-queue structure of TOE and the single DDR3 structure.

Firstly, to observe the law of transmission delay variation with update interval $t_{s}$, three observation points A, B, and C are set, where the update intervals are 85 us, 10 us, and 45 us, respectively, and the rest of the update intervals are varied randomly from 5 to 100 us.

As shown in Figure 18, the delays of both the double-queue structure and the single DDR3 structure of TOE fluctuate when the update time changes, and the delays of both overlap around the three peak points A, B, and C. In addition, the latency of the single DDR3 structure is also affected by the payload length. The longer the length is, the larger the DDR3 write data delay is, thus increasing the latency.

Take point A as an example. In this orange area, the application layer has not been releasing new capacity, so the direct query module interacts to obtain the result X > Y, and then the current packet takes the indirect channel. DDR3-stored data pile up and the delay increases to 2.2 us. Until the application layer releases a new capacity of 2000 bytes, the payload data of several stalled packets are read out, and the transmission gradually flows smoothly. When a new packet arrives, the direct channel query results in X ≤ Y and then takes the direct channel, and the delay gradually decreases to a low value of 600 ns.

Similarly, the delay at point B returns to 600 ns immediately after the end of the orange area. The delay at point C is high only after the end of the orange area in the third part because, in that region, the application layer still has spare capacity for TOE to transfer data, while at point C, the length to be transferred X > Y and the delay increases to 1.3 us as the application layer updates its capacity faster. TOE quickly sends the data to be transferred through DDR3, and eventually, it returns to the low value of 600 ns.

Secondly, to observe the variation of the transmission delay with the release space capacity $B_{s}$, an observation point D is set. As shown in Figure 19, the orange area is 1500 bytes and the rest is 2000 bytes. The transmission delay of the two TOE structures is more affected by the release space capacity compared to the payload length.

Taking point D as an example, as the space capacity release in the orange area becomes less as packets keep coming in, X is always greater than Y. Then, the DDR3 of the indirect channel is used. The TOE needs to wait for the time $t_{s}$ to release its capacity until Y is not zero. At this time, the packets stuck in the DDR3 are gradually sent out, so the delay becomes smaller. However, TOE is still receiving new packets until Y is zero again, and the latency increases again, showing a “sawtooth” pattern. As more data pile up in the DDR3, the latency grows significantly until it reaches a peak.

After the D point, the released space capacity $B_{s}$ returns to normal, and Y starts to accumulate from zero. Until all the data to be transmitted by the DDR3 are completed, a new packet arrives, the direct channel query results in X ≤ Y, which takes the direct channel, and the delay gradually decreases to a low value of 600 ns.

The experimental results show that the transfer latency of TOE is affected by the load length, DDR3 write characteristics, and processing parameters (update interval and released space capacity) of the application layer. The fluctuation is more obvious when the rate is slower due to the change of the application layer transfer parameters. However, in general, the transmission latency performance of TOE’s double-queue storage structure is superior to that of the single DDR3 structure.

### 5.3. Maximum Number Experiment of TCP Sessions

To test the 1024 TCP sessions supported by the TOE proposed in this paper, the experiment uses the TCP/UDP performance testing tool as shown in Figure 20, which enables Host A to send 1024 TCP chain-building requests to the TOE at 50 ms intervals. The source port is incremented and the source IP address is 192.168.116.20. The destination port is set to 8088 and the destination IP address is 192.168.116.1. At the same time, the TCP packet transmission of 1024 TCP sessions is observed by the packet capture test software named Wireshark. The experimental results are shown in Figure 21 and Figure 22.

Figure 21 is the result of Wireshark’s packet capture; it can be seen that Host A sends chain-building SYN packets to TOE, TOE replies with SYN + ACK response packets, and Host A replies with ACK packets. Thus, it completes the three handshake processes required for TCP single-session chain-building. The chain-building process is repeated, indicating that the chain-building function of the TOE is normal.

Figure 22 shows the total captured conversation statistics of all TCP sessions, and from the green circle, there are 1024 TCP sessions working properly in total. The experimental results show that the TOE in this paper can support 1024 TCP sessions.

### 5.4. Interactive Query Mechanism Experiment

In order to verify the slicing process after the interactive query in the indirect query module, we designed the interactive query mechanism experiment as shown in Figure 23.

To observe the relationship between the probability of space sufficiency in the application layer and the slicing situation, we set the probability of space sufficiency to vary from 25% to 50% randomly, and the packet length to vary from 64 to 512 bytes. The number of concurrent sessions is varied to observe whether the packet is effectively sliced.

As can be seen in Figure 23a, from the sampled eight data frames of different lengths, four data frames are not sliced and the rest of the data frames are sliced due to the insufficient amount of data Y available for transmission at the application layer. The total amount of sliced data equals the payload length of the data frames. Similarly, (b) and (c) are also sliced according to the value of Y, and the total length remains equal to the payload length.

When the probability of space sufficiency is reduced to 25%, the sampled data frames of session index 1 are sliced and increased to seven, thus showing that the more severe the application layer blockage situation is, the more data are sliced. Similarly, observing (e) and (f) reveals that the total amount of sliced data frames for session indices 2 and 3 is still equal to the payload length of the data frame.

The experimental results show that the interactive query slicing mechanism designed in this TOE is accurate and has no data loss. In the case of a successful current connection establishment, the more severe the application layer blockage is, the fewer slicing errors will occur. Such a mechanism can well ensure the stability and high performance of TCP packet data transmission and avoid data transmission blockage caused by insufficient space for a single concurrent session.

### 5.5. TOE Reception Performance Experiment

To test the reception rate performance of network data frames within the TOE, a Python program is written to send a 100 GB size packet to the TOE from Host A. We observe the sender-side rate of Host A of the task manager to derive the reception rate of the TOE. The results are shown in Figure 24, the TOE’s reception performance is 9.5 Gbps, which is in line with the 10-gigabit rate.

### 5.6. TOE Resource Analysis

This paper evaluates the related resources based on the Xilinx FPGA VC709 development kit with default synthesis options, where FF denotes flip-flop resources, LUT denotes lookup table resources, and BRAM denotes programmable memory module resources. Table 2 shows the resource utilization of TOE, which supports 1024 TCP sessions; each TCP session has a 2 Mb send/receive buffer. To facilitate pipeline implementation, we use some registers and buffers between some modules. From Table 2, we can see that a total of 51,591 LUTs, 69,031 FFs, and 363 BRAMs are used.

## 6. Conclusions

In this paper, we further investigate and optimize the storage structure of all data inside the TCP stack, and propose a 10 G TCP/IP offload engine based on FPGA. We construct a TOE reception transmission delay theoretical analysis model, optimize the delay by using a double-queue storage structure, and verify the correctness of the model by parameter analysis experiments.

After board-level testing and verification, this TOE has a reception rate of 9.5Gbps, a minimum system latency of 600 ns, and supports 1024 TCP sessions. When transmitting TCP data frames with a payload length of 1024 bytes, the performance is improved by at least 55.3% compared to implementation approaches in the same condition and is only 3.2% of the software implementation approaches.

In the future, we will focus on improving the functionality of the TCP protocol, extending the maximum number of concurrent TCP sessions, and optimizing the transmission delay on the sender side. New TOE modules will also be developed for more application scenarios to maximize efficiency.

## Figures and Tables

**Figure 1 sensors-23-04690-f001:**
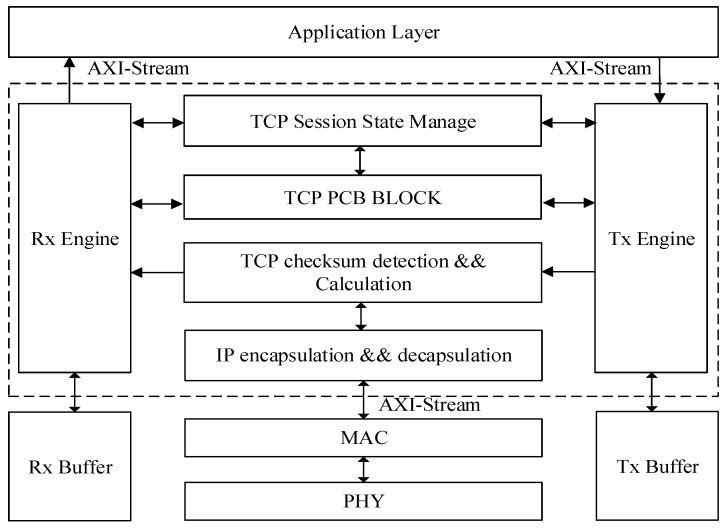
The TOE framework architecture.

**Figure 2 sensors-23-04690-f002:**
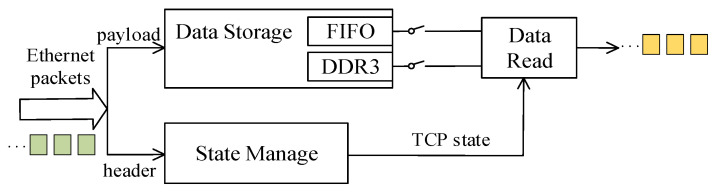
The TOE reception principle.

**Figure 3 sensors-23-04690-f003:**
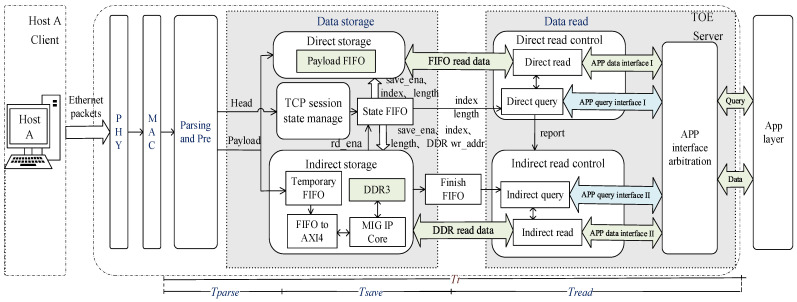
Proposed FPGA-based TOE reception transmission model structure. Parsing and Pre: parsing and preprocessing. APP layer: application layer.

**Figure 4 sensors-23-04690-f004:**
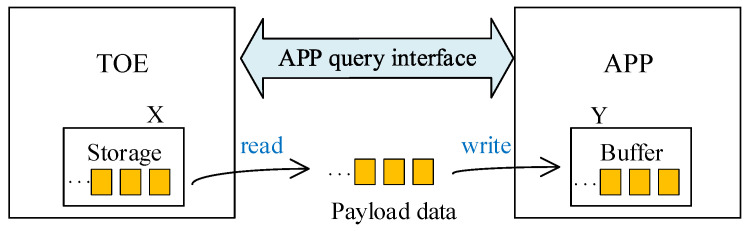
Query interaction and data transmission process between TOE and the application layer.

**Figure 5 sensors-23-04690-f005:**
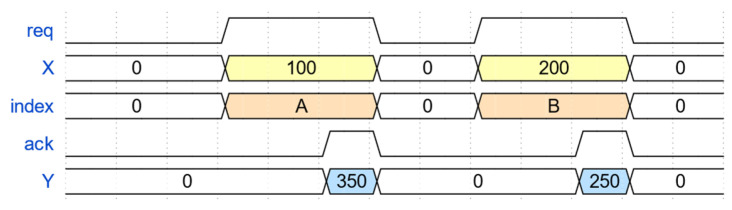
APP query interface signal diagram.

**Figure 6 sensors-23-04690-f006:**
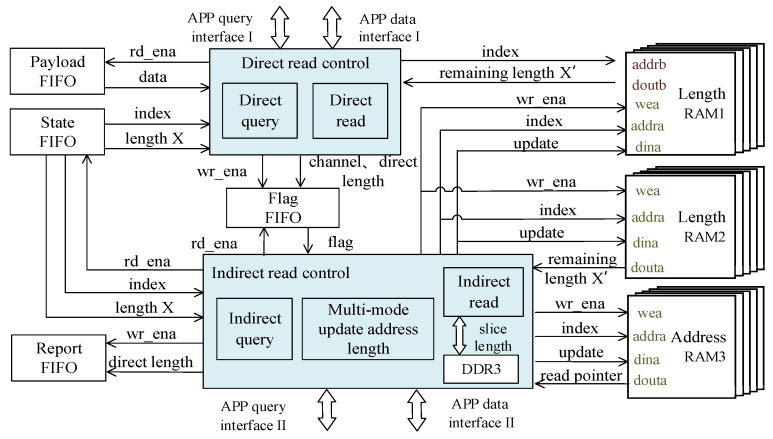
Transmission scheduling strategy diagram.

**Figure 7 sensors-23-04690-f007:**
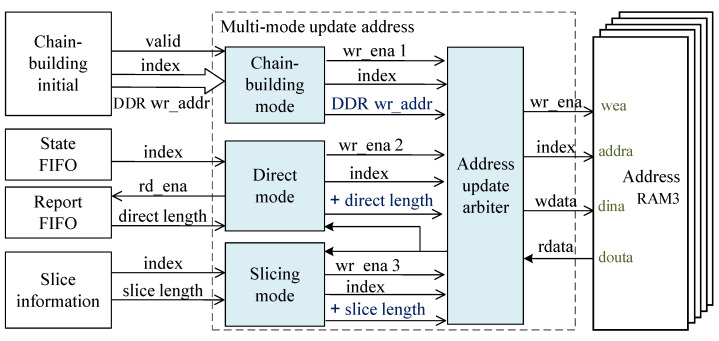
Multi-mode update address method structure diagram.

**Figure 8 sensors-23-04690-f008:**
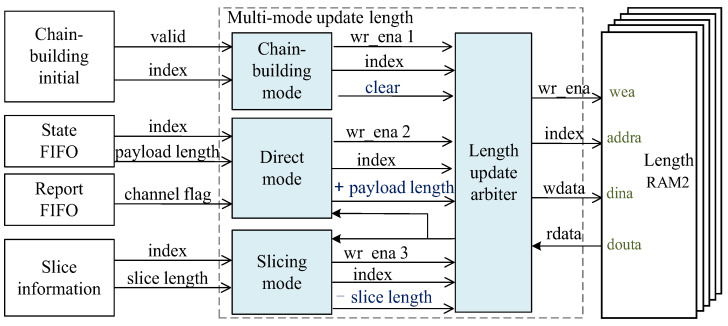
Multi-mode update length method structure diagram.

**Figure 9 sensors-23-04690-f009:**
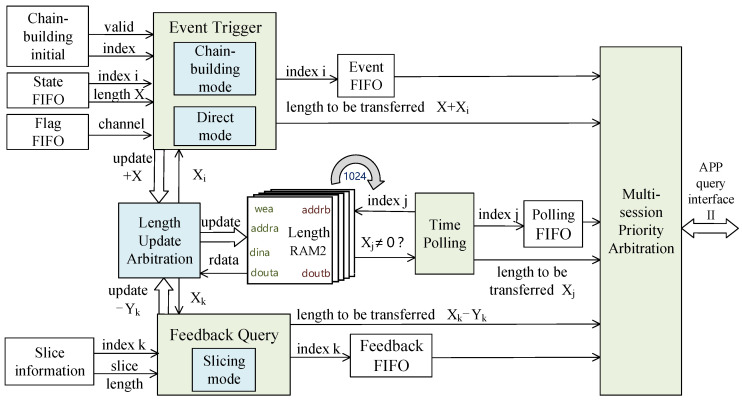
Multi-session priority arbitration method structure diagram.

**Figure 10 sensors-23-04690-f010:**
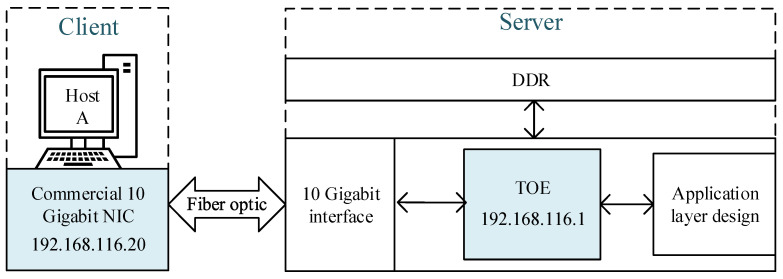
Test connection topology diagram.

**Figure 11 sensors-23-04690-f011:**
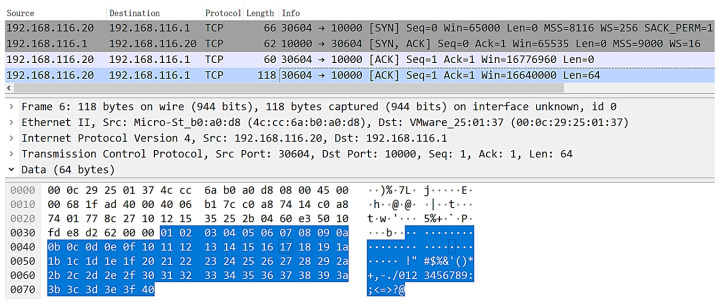
TOE received TCP data frame packet capture results.

**Figure 12 sensors-23-04690-f012:**
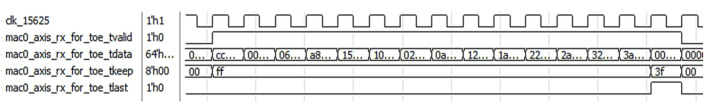
Input interface signal format.

**Figure 13 sensors-23-04690-f013:**
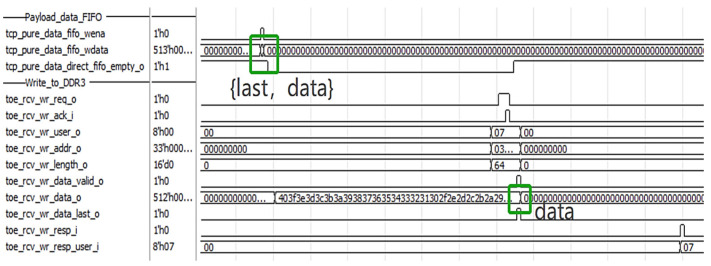
Data storage signals.

**Figure 14 sensors-23-04690-f014:**
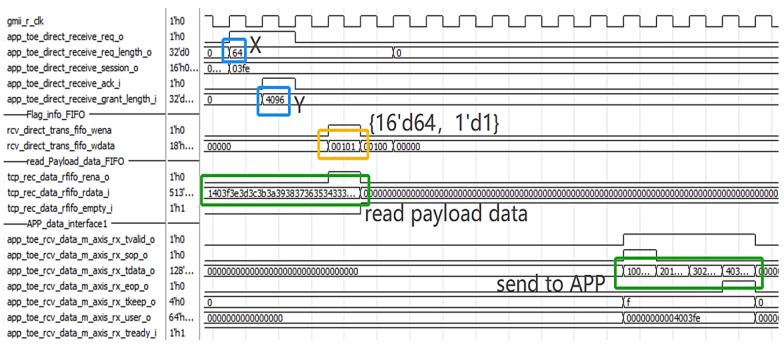
Direct reading of control signals during data reading.

**Figure 15 sensors-23-04690-f015:**
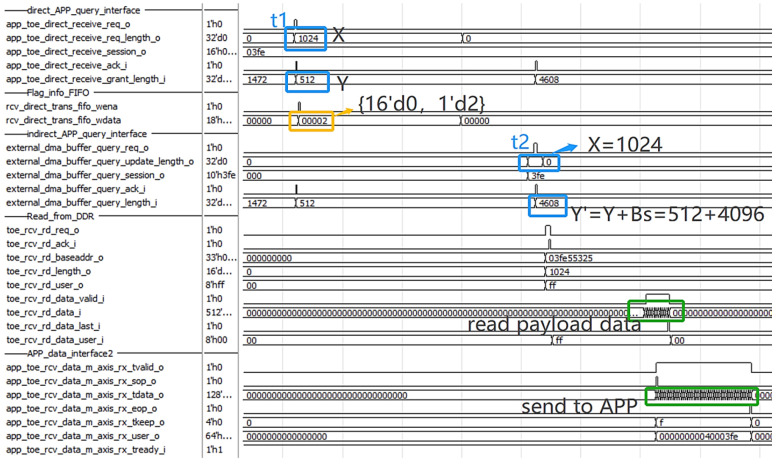
Indirect reading of control signals during data reading.

**Figure 16 sensors-23-04690-f016:**
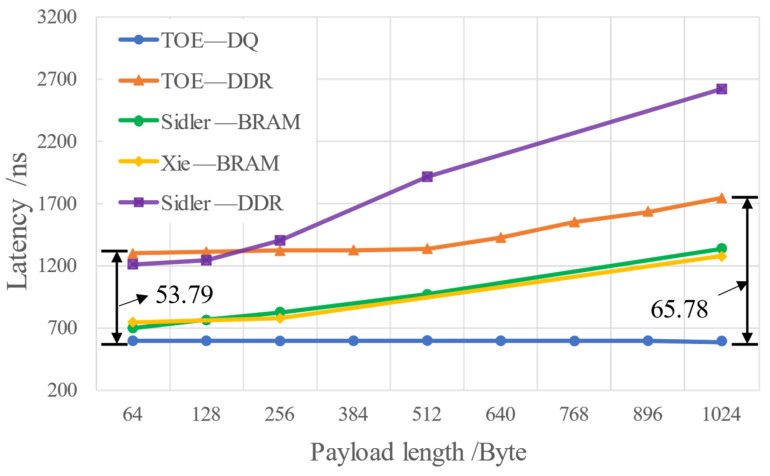
The transmission latency performance of different hardware approaches with different payload lengths. TOE—DQ: double-queue storage structure TOE. TOE—DDR: single DDR3 storage structure TOE. Sidler—BRAM: BRAM storage structure from [9]. Xie—BRAM: BRAM storage structure from [21]. Sidler—DDR: single DDR3 storage structure from [8].

**Figure 17 sensors-23-04690-f017:**
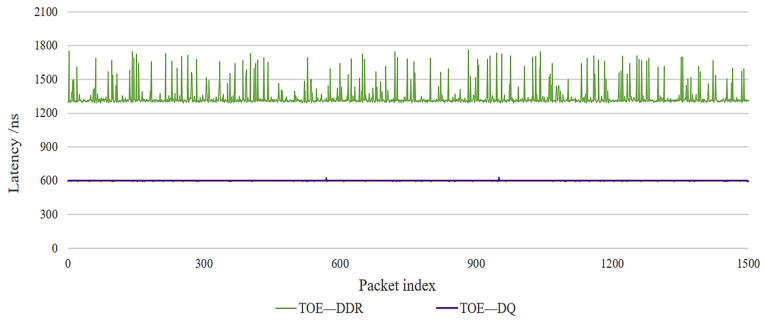
Comparison of TOE transmission delay with different indexes of packets. TOE—DDR: single DDR3 storage structure TOE. TOE—DQ: double-queue storage structure TOE.

**Figure 18 sensors-23-04690-f018:**
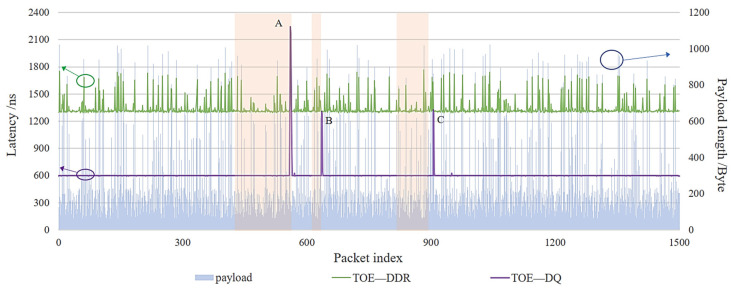
Comparison of TOE transmission delay between TOE’s double-queue structure and DDR3 structure with different update time interval $t_{s}$. payload: different payload lengths, its value is in the right vertical coordinate (Blue circles and blue arrows point in the direction). TOE—DDR: single DDR3 storage structure TOE, its value is in the left vertical coordinate (Green circles and green arrows point in the direction). TOE—DQ: double-queue storage structure TOE, its value is in the left vertical coordinate (Purple circles and purple arrows point in the direction).

**Figure 19 sensors-23-04690-f019:**
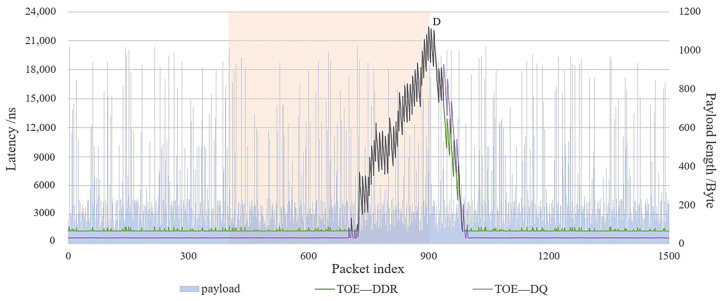
Comparison of TOE transmission delay between TOE’s double-queue structure and DDR3 structure with different the release space capacity $B_{s}$. payload: different payload lengths, its value is in the right vertical coordinate. TOE—DDR: single DDR3 storage structure TOE, its value is in the left vertical coordinate. TOE—DQ: double-queue storage structure TOE, its value is in the left vertical coordinate.

**Figure 20 sensors-23-04690-f020:**
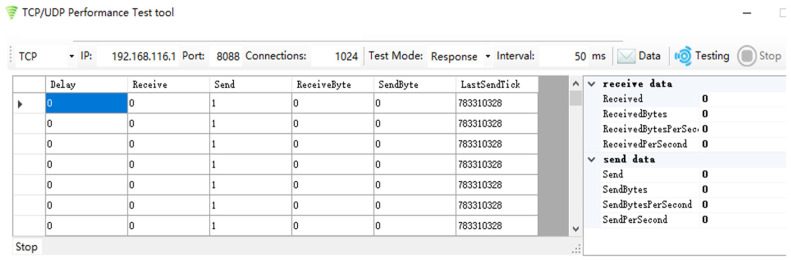
TCP/UDP performance testing tool settings.

**Figure 21 sensors-23-04690-f021:**
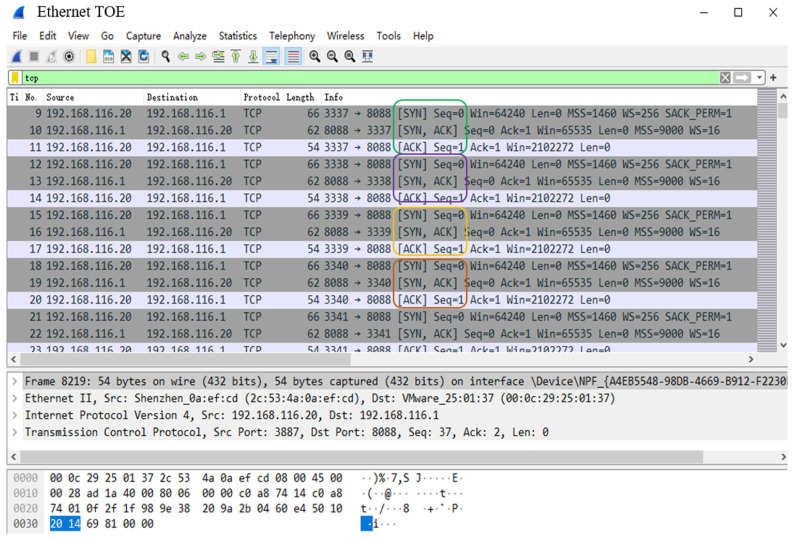
Packet capture results of Wireshark. These frames with different colors represent chain-building operations for different TCP sessions, and the three packets in the same colored box are a set of three handshake processes.

**Figure 22 sensors-23-04690-f022:**
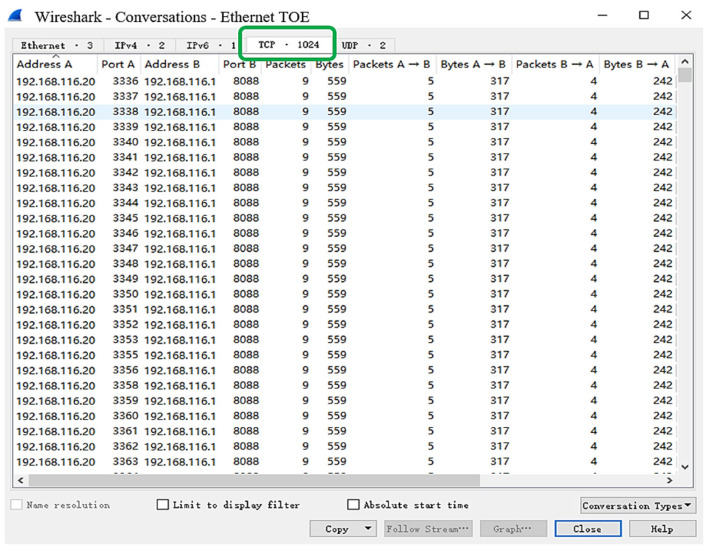
Conversation statistics of all TCP sessions captured by Wireshark.

**Figure 23 sensors-23-04690-f023:**
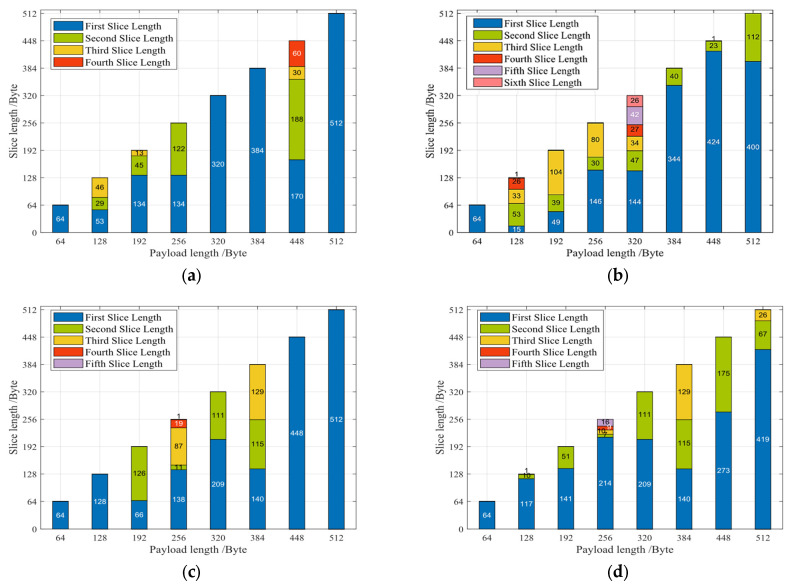
The slicing results of multi-session interactive query mechanism. (**a**) Slicing results for session index 1 with 50% sufficient probability. (**b**) Slicing results for session index 1 with 25% sufficient probability. (**c**) Slicing results for session index 2 with 50% sufficient probability. (**d**) Slicing results for session index 2 with 25% sufficient probability. (**e**) Slicing results for session index 3 with 50% sufficient probability. (**f**) Slicing results for session index 3 with 25% sufficient probability.

**Figure 24 sensors-23-04690-f024:**
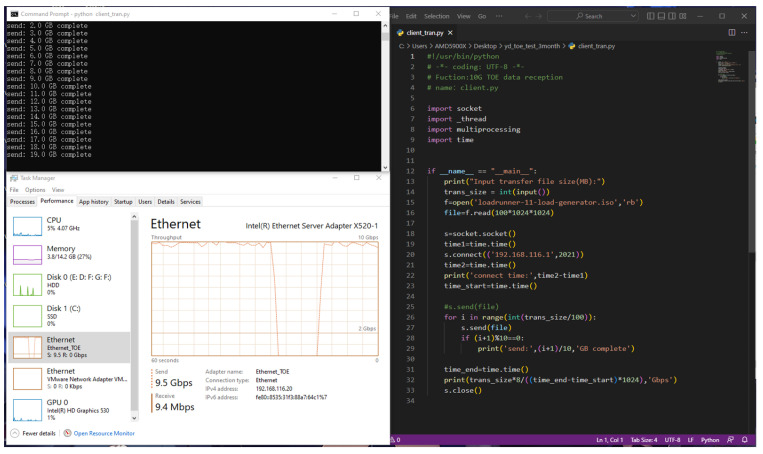
The experimental results of TOE reception data rate performance.

**Table 1 sensors-23-04690-t001:** System configuration parameters.

System Parameter	Category	Value	Unit
Data clock frequency	MAC	156.25	MHz
	TOE	200	
	APP	200	
Data bit width	MAC	64	
	TOE	512	bits
	APP	128	
IP address	Host A	192.168.116.20	/
	TOE	192.168.116.1	
Port number	Host A	30604	/
	TOE	10000	

**Table 2 sensors-23-04690-t002:** TOE resource consumption.

Resource	Utilization	Available	Utilization%
LUT	51,591	433,200	11.91
FF	69,031	866,400	7.97
BRAM	363	1470	24.69

## Data Availability

Publicly not supported at this time.

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
