# Peer review of "Low Latency TOE with Double-Queue Structure for 10Gbps Ethernet on FPGA"

_sensors, 2023, doi:10.3390/s23104690_

Round 1

Reviewer 1 Report

This paper studies the optimization of the storage structure of all data inside the TCP stack and proposes a 10G TCP/IP offload engine based on FPGA. A TOE reception transmission delay theoretical analysis model is constructed and experimentally verified. My observations and comments on this work are :

1. The delays are modeled as time-varying or fixed. Please elaborate on this point.

2. Elaborate Fig. 3.

3. There is a need for a comparative study with the existing approach to showcase the efficiency as well. This will undoubtedly attract the audience for the proposed method.

4. Can the contents in Fig. 17 be changed into English to have a better understanding?

5. Introduction needs refinement from the literature survey viewpoint.

Reviewer 2 Report

The methodology and experimental design are well-constructed and clearly explained. However, there are some areas where the writing could be improved to enhance the clarity and readability of the manuscript.

1. My main concern with the manuscript is the lack of a detailed introduction to the TOE (TCP/IP Of-fload Engine) framework. It is important to provide a comprehensive introduction to the TOE framework, which should be self-contained and fully integrated into the manuscript.

2. I find that the discussion of related work is somewhat superficial. While you have included a list of references, the descriptions are brief and do not provide enough detail to demonstrate the significance of the work. Furthermore, some of the cited references are quite old, which raises questions about the relevance and applicability of the proposed method to current research. Considering the importance of the comparison of different methods for demonstrating the novelty and contribution of your research, I strongly recommend that you consider adding a dedicated "Related Work" section to your manuscript.

3. The author should follow secure design principles when using TOE technology, despite the manuscript's goal of minimizing transmission latency. Brief security analysis should be added to the design section, including security analysis of protocols and algorithms, secure hardware design, and security controls for interfaces and APIs. If the designed method has security vulnerabilities or needs improvement, it should be clearly pointed out.

I recommend that you carefully proofread the manuscript for errors in grammar, spelling, and punctuation. These types of errors can detract from the overall quality of the manuscript and make it more difficult for readers to follow.

Round 2

Reviewer 1 Report

The responses are well-addressed. Thanks!

Author Response

Thanks for the recognition of our contributions.

Reviewer 2 Report

I am glad that authors revised this paper based on comments. This paper can be accepted.

However, one problem is how to deal with small number and large number TCP sessions in section 3,4,5 and 5? In abstract, there is “TOE supports 1K TCP sessions”. In conclusion, there is “supports 1024 concurrent TCP sessions”. Please give some revision.
